# Gene–Smoking Interaction Analysis for the Identification of Novel Asthma-Associated Genetic Factors

**DOI:** 10.3390/ijms241512266

**Published:** 2023-07-31

**Authors:** Junho Cha, Sungkyoung Choi

**Affiliations:** 1Department of Applied Artificial Intelligence, College of Computing, Hanyang University, 55 Hanyang-daehak-ro, Sangnok-gu, Ansan 15588, Republic of Korea; chajunho822@hanyang.ac.kr; 2Department of Mathematical Data Science, College of Science and Convergence Technology, Hanyang University, 55 Hanyang-daehak-ro, Sangnok-gu, Ansan 15588, Republic of Korea

**Keywords:** asthma, gene–environment interactions, Korean Genome and Epidemiology Study, genome-wide association study, smoking status

## Abstract

Asthma is a complex heterogeneous disease caused by gene–environment interactions. Although numerous genome-wide association studies have been conducted, these interactions have not been systemically investigated. We sought to identify genetic factors associated with the asthma phenotype in 66,857 subjects from the Health Examination Study, Cardiovascular Disease Association Study, and Korea Association Resource Study cohorts. We investigated asthma-associated gene–environment (smoking status) interactions at the level of single nucleotide polymorphisms, genes, and gene sets. We identified two potentially novel (*SETDB1* and *ZNF8*) and five previously reported (*DM4C*, *DOCK8*, *MMP20*, *MYL7*, and *ADCY9*) genes associated with increased asthma risk. Numerous gene ontology processes, including regulation of T cell differentiation in the thymus (GO:0033081), were significantly enriched for asthma risk. Functional annotation analysis confirmed the causal relationship between five genes (two potentially novel and three previously reported genes) and asthma through genome-wide functional prediction scores (combined annotation-dependent depletion, deleterious annotation of genetic variants using neural networks, and RegulomeDB). Our findings elucidate the genetic architecture of asthma and improve the understanding of its biological mechanisms. However, further studies are necessary for developing preventive treatments based on environmental factors and understanding the immune system mechanisms that contribute to the etiology of asthma.

## 1. Introduction

Asthma is one of the most prevalent chronic respiratory diseases, with a worldwide burden associated with loss of life and overall morbidity [1,2]. It affected more than 330 million people in 2019 [3,4] and results in approximately 400,000 deaths annually [3,5,6]. The general symptoms of asthma include wheezing, coughing, chest tightness or pain, and shortness of breath [7]. Asthma can lead to respiratory distress, obstructive sleep apnea, psychological disorders, seizures, and other complications [8,9,10].

Asthma is a complex disease caused by a combination of multiple genetic and environmental factors [11,12]. In terms of genetic factors, heritability estimates range between 35 and 95% [13], with those based on twin studies ranging between 50 and 90% [14,15]. The first genome-wide association study (GWAS) for asthma was published in 2007 [16] and showed that genetic variants on chromosome 17q21.1 were associated in cis with transcript levels of *ORMDL3*. Numerous GWASs for asthma followed, and more than 140 susceptibility loci have been identified [17,18,19,20,21,22,23,24,25]. The identified variants in common risk alleles describe only a small proportion of asthma heritability [17,18,19]. The “missing heritability” is due to rare structural variants, such as copy number variants and insertion–deletion mutations (INDELs) [26,27]. However, exome sequencing revealed that the significance of rare variants in the development of asthma remains unknown [28]. Furthermore, several studies on asthma did not provide strong evidence that structural variants have a role in asthma susceptibility [29,30,31]. Further, the “missing heritability” is due to an overestimation of the role of genetics in asthma development [26,27,32]. Regarding the findings of twin studies, monozygotic twins have been found not only to be genetically identical when compared with dizygotic twins but also to have an environment identical to that of dizygotic twins [33]. Lastly, the “missing heritability” could be due to gene–gene and gene–environment interactions [27]. Systems biology applications that account for highly interconnected molecular networks have been proposed to better understand gene–gene interactions; however, the missing heritability of complex diseases still remains poorly understood. In contrast, gene–environment interaction studies have steadily progressed, and several environmental causes of asthma and allergic diseases have been identified [12,34,35,36,37,38]. Smoking status is a major environmental factor associated with the development of asthma [36,37]. Cigarettes contain nicotine and more than 4000 compounds that bind to and chemically alter or damage DNA [39]. For instance, genetic variants in 17q21 and 20p13-p12 were found to be significantly associated with asthma in patients exposed to smoking [40,41,42,43]. Furthermore, several GWASs have investigated the interactions between single nucleotide polymorphisms (SNPs) and smoking status [44,45,46]. GWAS have made considerable efforts in better understanding missing heritability. However, due to the small number of replicates, insufficient biological plausibility, and lack of statistical power, missing heritability remains poorly understood [47].

Therefore, to further understand the genetic architecture of asthma, we aimed to discover the effects of smoking status on asthma by analyzing genetic information from the Korean Genome and Epidemiology Study (KoGES) [48]. We evaluated gene–smoking interactions using SNP, gene, and gene-set analyses. To prioritize regulatory variants from the GWAS loci, we implemented linkage disequilibrium analyses and publicly available functional annotation tools: combined annotation-dependent depletion (CADD) [49], deleterious annotation of genetic variants using neural networks (DANN) [50], and RegulomeDB [51]. The study expands our knowledge of the genetic contribution to the development of asthma, thereby providing insight into asthma susceptibility and disease mechanisms.

## 2. Results

### 2.1. General Characteristics of Study Participants

As shown in Table 1, a total of 66,857 participants were enrolled in the current study. The general characteristics of the participants for asthma analysis were described by the defined groups “case” and “control” in the Health Examination Study (HEXA), Cardiovascular Disease Association Study (CAVAS), and Korea Association Resource Study (KARE) cohorts of the KoGES consortium. The HEXA cohort included 975 cases with a mean age of 55.4 ± standard deviation (SD) 8.4 years and 57,479 controls with a mean age of 53.8 ± 8.0 years. The CAVAS cohort included 95 cases and 2908 controls with a mean age of 57.9 ± 7.8 and 55.4 ± 7.8 years, respectively. The KARE cohort included 112 cases and 5308 controls with a mean age of 53.3 ± 7.9 and 51.5 ± 8.5 years, respectively. Age was significantly associated with asthma in all three cohorts (*p* < 0.005, *t*-test). The average body mass index (BMI) values of patients with asthma (cases) and controls were 24.3 ± 3.2 and 23.9 ± 2.9 kg/m^2^ in the HEXA cohort (*p* = 0.0002, *t*-test), 25.5 ± 3.4 and 24.5 ± 3.0 kg/m^2^ in the CAVAS cohort (*p* = 0.0002, *t*-test), and 25.0 ± 3.5 and 24.6 ± 3.0 kg/m^2^ in the KARE cohort (*p* = 0.1536, *t*-test), respectively. The male-to-female ratio (SEX) of cases and controls was 29.0% and 71.0% in the HEXA cohort, 38.9% and 61.1% in the CAVAS cohort, and 34.8% and 65.2% in the KARE cohort, respectively. The SEX was significantly associated with asthma in the HEXA and KARE cohorts. The allergy (ALLER) status was significantly associated with asthma in all three cohorts (*p* < 0.0001, chi-square test). To investigate the effect of smoking status on asthma, we performed an association analysis, using a logistic regression model adjusted for age, sex, BMI, and ALLER status. A significant association between smoking status and asthma was found in both the HEXA (*p* = 0.0003, logistic regression) and KARE cohorts (*p* = 0.0129, logistic regression). However, such an association was not found in the CAVAS cohort (*p* = 0.6627, logistic regression).

### 2.2. GWAS Results

We performed gene–environment (smoking status) interaction analysis (GWAS) to identify the genetic factors (7,060,677 SNPs from 66,857 subjects from the HEXA, CAVAS, and KARE cohorts) associated with asthma, which were analyzed after genotype quality control (Figure 1). The *p*-values for the gene–environmental interactions factor were determined using the PLINK software (v1.90) [52], and the “--interaction” option. In Appendix A, the quantile–quantile (Q–Q) plot showed no inflation of *p*-values. The lambda GC (*λ*_GC_, genomic control) [53] in our analysis was 0.98. Appendix A shows the Manhattan plot of the *p*-values in the GWAS gene–environment interaction analysis for asthma, where the horizontal red line denotes the threshold for a 0.05 genome-wide significance level by a Bonferroni correction of 7.08 × 10^–9^. Table 2 summarizes the top 40 SNPs with *p*-values less than the nominal significance level of 1.00 × 10^–5^. Notably, the most significant SNP–smoking status interacting signal was *rs77079226* (OR = 3.083, 95% CI 1.985–4.788, *p* = 5.37 × 10^−7^), located in *TMEM74* on chromosome 8. This study anticipated a strong interaction between *rs77079226* and smoking status in asthma, indicating the deleterious effect of *rs77079226*(C). Conversely, in the case of *rs2292731* (OR = 0.637, 95% CI 0.523–0.775, *p* = 6.68 × 10^−6^), we found an protective interaction effect between *rs2292731* and smoking status on the risk of asthma. Furthermore, *KDM4C* [54,55,56,57] and *MMP20* [58,59,60] were previously known to be associated with asthma.

### 2.3. Gene Analysis and Gene-Set Analysis

Using multi-marker analysis of genomic annotation (MAGMA) v1.08 [61] with the SNP-wise mean model integrated into the Functional Mapping and Annotation (FUMA) v1.4.0 platform (https://fuma.ctglab.nl/ (accessed on 1 June 2023)) [62], all SNPs were mapped to 18,792 genes. Appendix A shows the Q−Q plot, which confirmed that there was no inflation in the test statistics (*λ*_GC_ = 0.98), and the Manhattan gene analysis plot. The glycerophosphodiester phosphodiesterase domain containing 1 (*GDPD1*) gene was found to have the lowest *p*-value, 5.75 × 10^−5^; however, the *p*-value was not statistically significant (*p*-value < 2.66 × 10^−6^ (0.05/18792)). The top 20 genes with a *p*-value < 1.00 × 10^−3^ are listed in Appendix A. Among these, *ADCY9* [63,64], *MYL7* [65], and *DOCK8* [66,67,68] have been previously reported to be associated with asthma. Gene-set analysis was performed using MAGMA integrated into FUMA, and the gene-set *p*-values were computed using the gene-based *p*-value for gene ontology (GO) terms, obtained from the Molecular Signatures Database (MsigDB) v7.0 (Gene Set Enrichment Analysis, UC San Diego, https://www.gsea-msigdb.org/gsea/msigdb/index.jsp (accessed on 1 June 2023)) [69,70,71]. In total, 9988 gene sets from the GO database (7343 GO biological processes (BPs), 1001 GO cellular components (CCs), and 1644 GO molecular functions (MFs)) were tested. The most significant GO process, which satisfied the Bonferroni-corrected significance level of *α* = 5.01 × 10^−6^ (*p* = 4.12 × 10^−6^), was the regulation of T cell differentiation in thymus (GO:0033081). The thymus is important to the immune system and contributes to the etiology of asthma [72,73,74]. The top five gene sets for each of the three GO sub-ontologies are shown in Figure 2. Among the 15 identified gene sets, the following were previously reported to be associated with asthma: BP—T cell differentiation in thymus (GO:0033077) [74]; BP—mast cell activation (GO:0045576) [75,76]; BP—relaxation of smooth muscle (GO:0044557) [77]; CC—azurophil granule (GO:0042582) [78,79]; CC—mast cell granule (GO:0042629) [80]; CC—ciliary membrane (GO:0060170) [81]; MF—beta galactosidase activity (GO:0004565) [82]; and MF—galactosidase activity (GO:0015925) [82].

The regulation of T cell differentiation in the thymus (GO:0033081) is associated with nicotine as a main component of cigarettes via smoking [83,84,85]. In previous studies, nicotine exposure was found to affect the thymus, an important organ of the immune system, and to interfere with epithelial cell adhesion and growth [83,84]. Nicotine leads to T cell downregulation and maturation and to T helper (Th)1/Th2 imbalance and reduced immunity [83,84,86,87,88]. Nicotine also activates the nicotinic acetylcholine receptor (nAChR) and modulates thymocyte development [83]. T cell nicotinic responses were demonstrated to occur through the α7 subunit [84]. α7 nAChR activation triggers extracellular Ca^2+^ influx associated with upregulated Fas expression [89]. In a previous study, α7 nAChR was activated, and its levels increased concomitantly with caspase expression and apoptosis rate [90]. Moreover, α7 nAChR was shown to mediate nicotine pro-apoptotic effects on thymocytes by upregulating Fas expression and the Fas-mediated apoptotic pathway [91].

### 2.4. Functional Annotation

We applied follow-up bioinformatics analyses to two novel genes (*SETDB1* and *ZNF8*), identified in this study, and five previously reported genes (*KDM4C*, *DOCK8*, *MMP20*, *MYL7*, and *ADCY9*) using genome-wide functional prediction scores (CADD, DANN, and RegulomeDB). For the *SETDB1* gene, three variants had CADD scores > 10 or DANN scores > 0.7 and were associated with four transcription factor (TF)-binding sites. These TFs were previously reported to be associated with asthma (Table 3 and Appendix A) [92,93,94,95,96,97]. In the *ZNF8* gene, nine variants were associated with eight TF-binding sites (Appendix A), and four variants had DANN scores > 0.7 (deleterious, Table 3). Among the eight TFs, EZH2 and POLR2A have been associated with the pathogenesis of asthma [98,99]. Two *KDM4C* variants, ten *DOCK8* variants, one *MMP20* variant, two *MYL7* variants, and eight *ADCY9* variants had CADD, DANN, or RegulomeDB rank scores >10.0, >0.7, or ≤3, respectively (Appendix A). Among these, six variants were associated with TF-binding sites, and 31 TFs were associated with the pathogenesis of asthma (Table 3). However, we did not find significant results for *KDM4C* and *MYL7*.

## 3. Discussion

In this study, we conducted a gene–smoking interaction analysis to identify candidate genetic markers associated with asthma. We combined data from Korean multi-cohorts, (HEXA, CAVAS, and KARE) and identified two novel genes. We further contributed to understanding the functional associations of five previously identified asthma-associated genes. Using follow-up bioinformatics analyses, we also confirmed that two novel genes (*SETDB1* and *ZNF8*) and three previously reported genes (*DOCK8*, *MMP20*, and *ADCY9*) are potential asthma-associated genes. The *SETDB1* encodes a histone lysine N-methyl transferase; this important enzyme, involved in histone modification, is related to the *Th1* gene downregulation mechanism in asthma [72,73]. As shown in Figure 3, *SETDB1* causes epigenetic modifications through methylation of histone H3, which surrounds the DNA on the chromosome [72,146]. When histone H3 lysine 9 (H3K9) is methylated, deposition occurs in endogenous retroviruses and enhancer regions, which are necessary for *Th1* gene expression [72,73]. Downregulation of the *Th1* gene leads to overexpression of the *Th2* gene, which increases the number of Th2 cells, causing Th1/Th2 cell imbalance [73]. Th2 cells secrete IL-4 and IL-5 and increase IgE levels and eosinophil numbers, which can trigger inflammation, leading to asthma [146,147,148]. Moreover, SETDB1 can induce macrophage recruitment by promoting AKT/mTOR-dependent CSF-1 induction and secretion [149]. Increased macrophage levels due to overexpression of SETDB1 can cause inflammation by causing an increase in the pro-inflammatory cytokine IL-8 [150,151,152]. High expression of IL-4 by Th2 cell upregulation hyperstimulates M0 macrophages to differentiate into M2 macrophages [153]. Upregulated M2 macrophages overexpress monocyte chemotactic protein-1 (MCP-1), IL-8, IL-13, etc., which can increase the risk of inflammation and asthma onset [154,155,156]. In addition, *SETDB1* interacts with the asthma-related gene *SETDB2* [157]; the latter antagonizes the *KDM4C* gene to control H3K9 methylation [157], which affects the expression of *ORMDL3*, a well-known asthma-related gene [55,158]. Therefore, *SETDB2* dysregulation is associated with genomic instability and increased H3K9 methylation [157], leading to Th1/Th2 cell imbalance and asthma onset [73].

Furthermore, several studies have shown that nicotine from smoking can interact with *SETDB1* and induce asthma by causing Th1/Th2 cell imbalance [83,84,85,159,160]. In particular, these facts need to be investigated in conjunction with late-onset asthma (LOA) disease studies [161]. The phenotype of LOA is divided into two types according to the presence or absence of eosinophilic inflammation, Th2 and non–Th2-related LOA [162,163]. Th2-related LOA has been reported to be associated with sinusitis, nasal polyps, and sometimes aspirin-exacerbated respiratory disease (AERD) [164]. This type has recently been defined as uncontrolled asthma, often with severe symptoms from the onset [165]. Non-Th2-related LOA has been reported to be associated with gender, obesity, smoking, age, and corticosteroids and requires other treatment strategies, such as diet, macrolides, or smoking cessation [161,162]. In this study, nicotine exposure through smoking in adulthood degraded T cell regulation and maturation, induced Th1/Th2 imbalance and reduced immunity [83,84,86,87,88], and upregulated Fas expression and the Fas-mediated apoptosis pathway through α7 nAChR activation [91]. It has been suggested that non-Th2-related LOA can develop into severe Th2-related LOA. Further, this study suggested that the upregulation of *SETDB1* by genetic variants, such as *rs139189121*, *rs75406390*, and *rs59024312* (Table 3), can accelerate Th2-related LOA during nicotine exposure from smoking. *rs139189121*, *rs75406390*, and *rs59024312* are genic upstream transcript variants (Appendix A) located between exon 3 and exon 4 of *SETDB1*, which could influence *SETDB1* expression activity and transcription of the SETDB1 protein, resulting in a different asthma prognosis. Fine-mapping analysis revealed that *rs139189121*, *rs75406390*, and *rs59024312* were related to an increased risk of asthma when exposed to smoking (Appendix A), and it was confirmed through RegulomeDB chip-seq data (hg19). As a result, they could be bonded asthma-related proteins, such as AR [92], FOXA1 [93], GATA3 [94,95], and GATA6 [96,97] (Table 3). Therefore, further biological investigation is needed to elucidate whether *rs139189121*, *rs75406390*, and *rs59024312* can affect the expression of *SETDB1* and asthma prognosis.

Another novel gene identified in our study is *ZNF8* (Figure 4). *ZNF8* encodes the zinc finger protein 8, an important protein that interacts with and binds to MH1 and MH2 domains of the Smad1 protein and is related to *Smad1* suppression [166]. The main symptoms of asthma include reversible airway constriction, airway hyper-responsiveness, and chronic airway inflammation [167,168]. These symptoms are a complex process involving damage to the epithelial layer, hypertrophy and hyperplasia of the smooth muscles, and subepithelial fibrosis [169,170]. The key feature of subepithelial fibrosis is that human bronchial fibroblasts, derived from patients with asthma, show a characteristic transforming growth factor (TGF)-β1-induced fibroblast-to-myofibroblast transition (FMT) [166,170]. Antifibrotic TGF-β/Smad1 pathway activation is downregulated in fibroblasts; however, profibrotic TGF-β/Smad2/3 pathways are upregulated in patients with asthma [170]. Furthermore, the TGF-β/Smad1 and TGF-β/Smad2/3 pathways are antagonistic mechanisms [166,168,170]. Moreover, high expression of TGF-β1 is associated with eosinophil counts and asthma severity [171]. In asthmatic lungs, inflammatory cells express and secrete TGF-β1, which leads to an increase in eosinophils [171,172,173]. In particular, it has been reported that TGF-β1 is increased in chronic asthma and that eosinophils are the main source of asthma [172]. Thus, the ZNF8 protein, transcribed from *ZNF8,* interacts with Smad1, binds to the MH1 and MH2 domains of the Smad1 protein, and suppresses antifibrotic function [166]. Noteworthily, nicotine exposure from smoking increased *KDM4C* expression through epigenetic mechanisms [174]. These results suggest that the two novel genes (*SETDB1* and *ZNF8*) may play important roles in the etiology of asthma.

One of the main strengths of this study is that it utilizes an older cohort sample to provide a better LOA phenotype. In particular, among the two novel genes associated with asthma in this study, the association of *SETDB1* with LOA could be newly suggested through our analysis pipeline. Furthermore, this study focused on analyzing East Asian subjects, who have not been thoroughly investigated, unlike different ethnic groups. In this regard, we evaluated gene–smoking interactions using GWAS and gene analyses, revealing that genes, such as *DOCK8*, *MMP20*, and *ADCY9* were suggested by previous studies. Additionally, we conducted gene-set analysis using GO processes, finding that the regulation of T cell differentiation in the thymus (GO:0033081) is related to the pathogenesis of asthma.

Our study had several limitations. First, our sample size was relatively small compared to that of recent asthma GWAS cohorts, such as the Trans-National Asthma Genetic Consortium (TAGC) [175] and the UK Biobank cohort [176]. Additional asthma risk variants were identified as the sample size increased. Second, GWAS is subject to unbalanced binary traits; that is, there are fewer case samples than control samples. The imbalance could produce large type-I error rates in logistic regression results [177,178,179], and the significance of true causal variants may be overlooked [180,181]. Further machine learning approaches, such as the synthetic minority oversampling technique (SMOTE) [182] and cost-sensitive learning [183], are required to address for the imbalance-associated errors. Third, the lack of independent replication datasets did not allow us to validate all the top signal GWAS loci, genes, and pathways for asthma. Therefore, replication studies on asthma using Korean or East Asian cohorts are needed as follow-up studies. Fourth, bioinformatics analysis identified certain genetic factors and pathways related to asthma pathogenesis; however, the underlying biological mechanisms of these factors require further investigation, such as rare variants or tissue and cell-type enrichment analyses. Finally, GWAS functional analysis did not provide sufficient biological insight into the two novel genes associated with asthma. Further in vitro and vivo experimental studies related to the role of the two novel genes are needed to identify individuals at risk of asthma.

## 4. Materials and Methods

### 4.1. Study Participants

Epidemiologic and genetic data were collected by the KoGES consortium [48] and included three population-based study cohorts: HEXA (*n* = 61,568), CAVAS (*n* = 9715), and KARE (*n* = 8840). The KoGES consortium was a long-term follow-up cohort study conducted, in the Korean population, from 2001 to 2010. It was designed to investigate and assess genetic and environmental factors as correlates to the incidence of chronic diseases, such as type 2 diabetes, hypertension, and cancer [48]. The current study excluded subjects without Korean Biobank Array genotype data and those with missing epidemiologic information (SEX, AGE, BMI, ALLER, SMOKE, and asthma diagnosis). After applying the exclusion criteria, 66,887 participants were included in the study (HEXA, *n* = 58,434; CAVAS, *n* = 3003; KARE, *n* = 5420; Figure 1). Patients with asthma were defined based on their answer to the asthma history question, with participants being categorized as “case” or “control” if their answer was “yes” or “no,” respectively. The allergy status was defined based on the participants’ answer to the allergy history (rhinitis, atopy, allergic conjunctivitis, food allergy, etc.) question, with participants being categorized as “ALLER” or “non-ALLER” if their answer was “yes” or “no,” respectively. The smoking status was defined based on the participants’ answer to the smoking history question, with participants being accordingly categorized as “never-smokers,” “former-smokers,” or “current-smokers.” However, in this study, we gathered former-smokers and current-smokers in a single group (smokers).

All participants provided written informed consent to participate in the study. The participants’ genetic information was produced using the Korean chip through the Korea Biobank Array Project of the National Institutes of Health, Korea Centers for Disease Control and Prevention [184,185]. This study was approved by the Institutional Review Board of Hanyang University (IRB no. HYUIRB-202210-013).

### 4.2. Quality Control

Genomic DNA was genotyped using the Korean Biobank Array (Korean Chip, KORV1.1, Thermo Fisher Scientific, Waltham, MA, USA), designed by the Center for Genome Science (Korea National Institute of Health (KNIH)), based on the platform of the UK Biobank Axiom array and manufactured by Affymetrix [184,185]. SNP imputation was performed using the IMPUTE v2 software [186] with East Asian (Chinese and Japanese, 286 samples) 1000 Genomes Project Phase 3 data as a reference panel. Then, the PLINK software (version 1.90, Boston, MA, USA) [52] was used for SNP loci quality control. We excluded SNPs with missing genotype call ratio > 0.05, minor allele frequency (MAF) < 0.01, and Hardy–Weinberg equilibrium (HWE) *p*-value ≤ 10^−5^. After imputation and quality control, 7,060,677 SNPs were selected for association analysis.

### 4.3. Genome-Wide Association Analysis

Genome-wide association analysis was conducted using a logistic regression model, with the PLINK-G × E option, to test the interaction between genetic variants and environmental factors (smoking status). Single-SNP interaction effects were assessed using the following model:(1)$$logit\left(ASTH=1\right)=\beta_{0}+\beta_{1}SNP+\beta_{2}SMOKE+\beta_{3}SMOKE\times SNP+\beta_{4}SEX\;+\beta_{5}AGE+\beta_{6}BMI+\beta_{7}ALLER+\beta_{8}Groups1+\beta_{9}Groups2\;+\beta_{10}PC1+\beta_{11}PC2+\beta_{12}PC3+\beta_{13}PC4+\beta_{14}PC5\;+\beta_{15}PC6+\beta_{16}PC7+\beta_{17}PC8+\beta_{18}PC9+\beta_{19}PC10,$$
where *β*_0_, *β*_1_, *β*_2_, and *β*_3_ are the intercept value, effect sizes of the SNP, environmental factor, and interactions, respectively, and *β*_4_, *β*_5_, *β*_6_, and *β*_7_ denote coefficients of the covariate variables SEX, AGE, BMI, and ALLER, respectively. For the ASTH phenotype, the case group was denoted as 1, and the normal group was denoted as 0. The SNP was used as a marker to indicate the genetic information of each sample, and the genotype was assumed to be an additive genetic model (AA = 0, Aa = 1, and aa = 2, where “A” and “a” indicate the major and minor alleles, respectively). ALLER was categorized as either non-ALLER (coding = 0) or ALLER (coding = 1). SMOKE was classified into two groups: non-smokers (coding = 0) and smokers (former or current smokers; coding = 1). Moreover, as three integrated cohorts were analyzed in this study, we used covariates to adjust potential population stratifications in each cohort using dummy variables (HEXA: group 1 = 0, group 2 = 0; CAVAS: group 1 = 1, group 2 = 0; and KARE: group 1 = 0, group 2 = 1). Additionally, we conducted principal component (PC) analysis (PCA), which is the most widely used method to adjust for population stratification in GWASs [187]. We used the top 10 PC score that was calculated using the PLINK software (v1.90) and the “-- pca 10” option [52]. *β*_8_, *β*_9_, and *β*_9_–*β*_19_ denote the effect size of Groups 1, Groups 2, and PC1–PC10 values, respectively. To identify SNPs that interact with smoking status, we considered the following null hypothesis:(2)$$H_{0}:\;\beta_{12}=0$$

The *p*-values for the gene–environmental interaction factor, *β*_12_, that passed the Bonferroni correction threshold of 7.08 × 10^–9^ (0.05/7,060,677) were considered statistically significant.

### 4.4. Gene Analysis and Gene-Set Analysis

Gene and gene-set analyses were implemented in FUMA (v1.4.0) [62]. It used the input GWAS summary statistics to compute the *p*-value of gene-based and gene-set *p*-values, using the multi-MAGMA (v1.08) tool [61]. The MAGMA tool, with the multiple linear PC regression model [188], was used to identify multiple variant combined effects by incorporating linkage disequilibrium (LD) information among multiple variants within extended +/− 250 kb downstream or upstream of the gene [61]. We performed a MAGMA gene-based analysis using a reference panel from Phase 3 of the 1000 Genome Project East Asian data. For the gene-set analysis, we conducted the GENE2FUNC process to identify the putative biological mechanisms of the prioritized genes. The gene-set *p*-value was computed using a gene-based *p*-value for 9988 GO terms obtained from MsigDB (v7.0). Bonferroni correction was used to correct for multiple testing in both the gene and gene-set analyses.

### 4.5. Functional Annotation

Fine-mapping analysis was performed to prioritize candidate causal variants in the genomic regions surrounding the significant GWAS hits. For each GWAS hit, a SNP set consisting of genetic variants belonging to the same gene was selected and mapped to a significant SNP. Then, the LD among SNPs was computed using the Haploview software (v4.2) [189] to view the correlation structure. This is a commonly used functional annotation process considering LD to find the causal variant for the top SNP signals from the GWAS [190,191,192]. The most commonly used measures of LD are D′ and *r*^2^. D′ represents a statistic for estimating recombination differences, whereas *r*^2^ is the squared value of the correlation of the allelic states of two loci in the same gamete. Within the LD block containing a significant SNP, we extracted a list of adjacent SNPs with D′ > 0.9 or *r*^2^ > 0.8. Finally, functional annotation studies were performed to examine the relationship between asthma and the fine-mapping loci.

To annotate SNPs at our candidate loci, we conducted variant annotation (ANNOVAR) [193] (table_annovar.pl) using the hg19/GRCh37 reference genome and Ensembl gene (build 106). For intergenic SNPs, the gene boundaries overlap such that a single variant can be annotated to multiple genes. Therefore, the intergenic SNPs were mapped to the two closest upstream and downstream genes. All candidate SNPs were annotated using CADD (https://cadd.gs.washington.edu/ (accessed on 1 June 2023)), DANN (https://cbcl.ics.uci.edu/public_data/DANN/ (accessed on 1 June 2023)), and RegulomeDB scores (www.regulomedb.org/ (accessed on 1 June 2023)) to predict potential pathogenicity. For both coding and non-coding variants, CADD used more than 60 functional annotations to calculate deleterious scores. The raw CADD scores were then converted into PHRED-scaled scores ranging from 1 to 99, with higher values indicating more deleterious cases. For example, a PHRED-scaled score of 10 or greater indicates a raw CADD score in the top 10% of all possible reference genome SNPs. We also used a DANN to identify potentially functional SNPs using the VarSome prediction tool (v11.3.3, https://varsome.com (accessed on 1 June 2023)) [194]. The DANN score is a functional prediction scoring methodology with an output value in the range of 0 and 1, where higher values are more likely to indicate deleterious cases. RegulomeDB comprises datasets from the ENCODE Project Consortium, Gene Expression Omnibus, and other sources, totaling 962 datasets as well as published literature. RegulomeDB provides a SNP score that ranges from 1 to 6: the lower the SNP score, the stronger the regulatory association. The comparative characteristics of the RegulomeDB score categories are summarized in Table 4 [51].

## 5. Conclusions

This study demonstrated that two potentially novel genes (*SETDB1* and *ZNF8*) are associated with the development of asthma, based on gene–smoking interaction analyses using multiple cohorts of the KoGES consortium. Furthermore, the causal relationship between five genes (two potentially novel genes and three previously reported genes) and asthma was confirmed through follow-up bioinformatics analyses. Moreover, gene-set analysis revealed that the regulation of T cell differentiation in the thymus (GO:0033081) is significantly associated with the pathogenesis of asthma. Therefore, our findings indicate the importance of multi-levels analysis (GWAS loci, genes, pathways, and functional annotation) in gene–environment interaction studies, which may provide insight and an improved understanding of these complex traits. Our results reveal potential biological mechanisms that are involved in the pathogenesis of asthma. The results contribute to a greater understanding of asthma etiology and provide insights for further investigation.

## Figures and Tables

**Figure 1 ijms-24-12266-f001:**
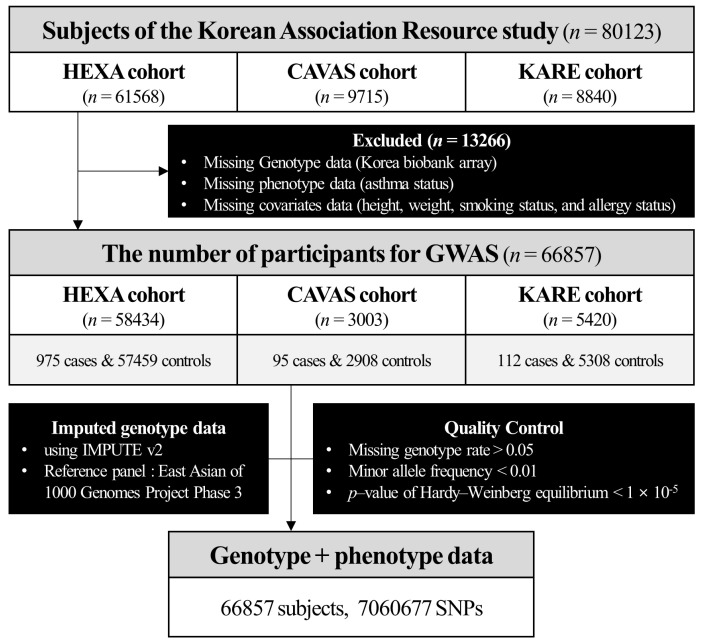
Schematic representation of the sample selection and study design.

**Figure 2 ijms-24-12266-f002:**
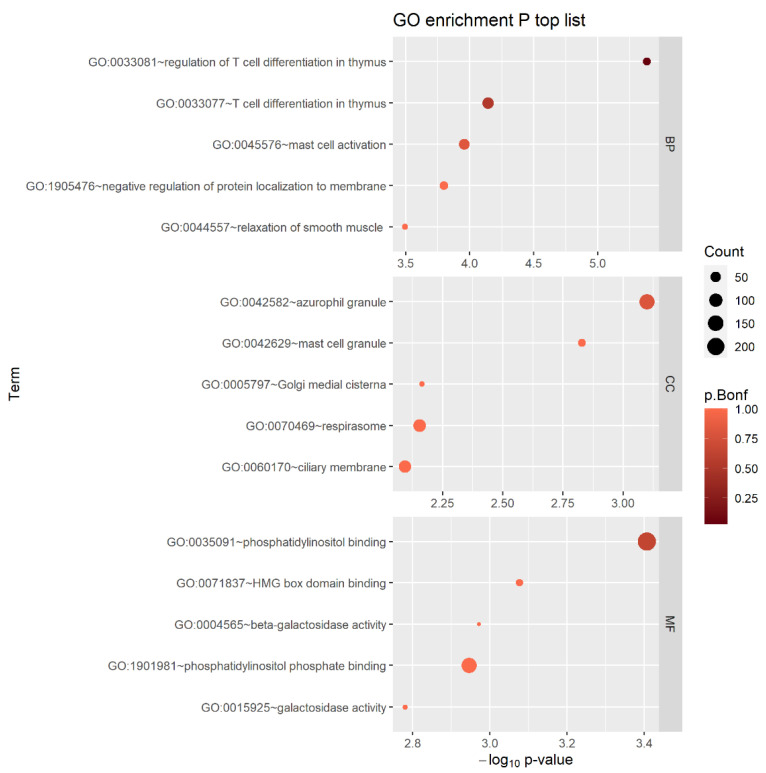
Bubble plot showing GO enrichment analysis (three sub-ontologies and top five enriched GO terms). Dot color corresponds to the Bonferroni-adjusted *p*-values and dot size to the number of genes mapped to the GO terms.

**Figure 3 ijms-24-12266-f003:**
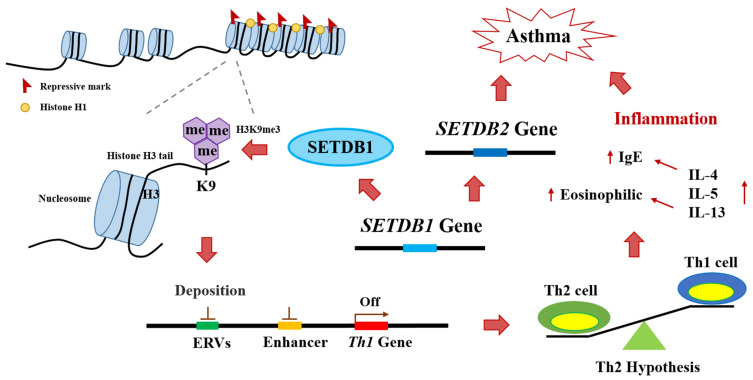
*SETDB1* plays an important role in asthma induction through histone 3 lysine 9 (H3K9) methylation. ERVs: Endogenous retroviruses.

**Figure 4 ijms-24-12266-f004:**
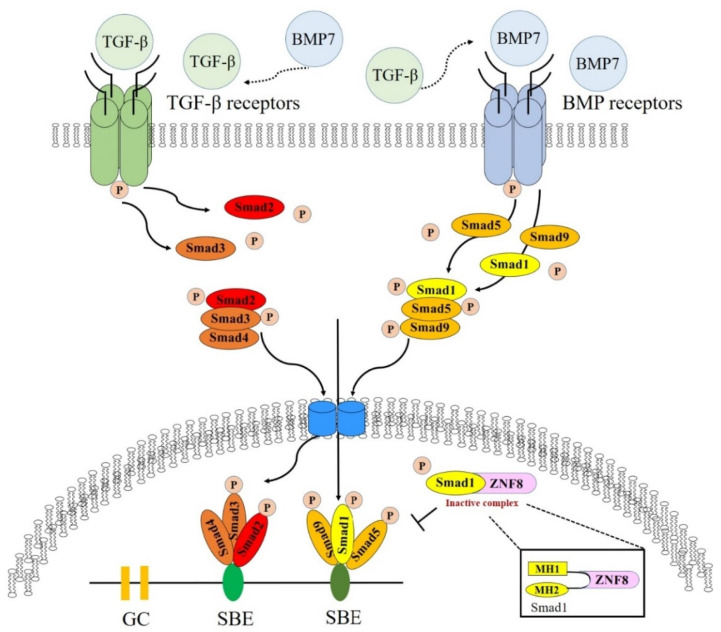
The *ZNF8* gene plays an important role in asthma induction by Smad1 interaction.

**Table 1 ijms-24-12266-t001:** General characteristics of participants from the HEXA, CAVAS, and KARE cohorts.

9	HEXA		*p*-Value	CAVAS		*p*-Value	KARE		*p*-Value
	Case (*n* = 975)	Control (*n* = 57,459)		Case (*n* = 95)	Control (*n* = 2908)		Case (*n* = 112)	Control (*n* = 5308)	
SEX			<0.0001			0.8325			0.0048
Male	283 (29.0%)	19,924 (34.7%)		37 (38.9%)	1164 (40.0%)		39 (34.8%)	2563 (48.3%)	
Female	692 (71.0%)	37,535 (65.3%)		58 (61.1%)	1744 (60.0%)		73 (65.2%)	2745 (51.7%)	
AGE (years) ^a^	55.4 ± 8.4	53.8 ± 8.0	<0.0001	57.9 ±7.8	55.4 ± 7.8	0.0025	53.3 ± 7.9	51.5 ± 8.5	0.0026
BMI (kg/m^2^) ^b^	24.3 ± 3.2	23.9 ± 2.9	0.0002	25.5 ±3.4	24.5 ± 3.0	0.0002	25.0 ± 3.5	24.6 ± 3.0	0.1536
ALLER status ^c^			<0.0001			<0.0001			<0.0001
Non-ALLER	727 (74.6%)	53,642 (93.4%)		74 (77.9%)	2695 (92.7%)		86 (76.8%)	5015 (94.5%)	
ALLER	248 (25.4%)	3817 (6.6%)		21 (22.1%)	213 (7.3%)		26 (23.2%)	293 (5.5%)	
Smoking status ^d, e^			0.0003			0.6627			0.0129
Non-smokers	721 (73.9%)	42,070 (73.2%)		72 (75.8%)	2123 (73.0%)		71 (63.4%)	3173 (59.8%)	
Smokers	254 (26.1%)	15,389 (26.8%)		23 (24.2%)	785 (27.0%)		41 (37.6%)	2135 (40.2%)	

^a^ Means ± standard deviation (SD); ^b^ body mass index (BMI); ^c^ allergy status (ALLER); ^d^ smoking status (Non-smoker: never smoker/Smoker: former smoker or current smoker); ^e^ the association between smoking status and asthma was analyzed by a logistic regression model adjusted for age, sex, BMI, and ALLER status.

**Table 2 ijms-24-12266-t002:** SNPs most significantly associated with smoking and the development of asthma (*p* ≤ 1.0 × 10^–5^).

CHR ^a^	rsID ^b^	Alt ^c^	MAF ^d^	OR	95% CI	STAT	*p*-Value	Location	Nearest Gene
8	*rs77079226*	C	0.034	3.083	1.985–4.788	5.012	5.37 × 10^–7^	Downstream transcript	*TMEM74*
10	*rs17153428*	A	0.297	1.618	1.329–1.970	4.791	1.66 × 10^–6^	Intron	*RP11-383C5.4*
22	*rs4823536*	G	0.103	2.013	1.511–2.682	4.781	1.74 × 10^–6^	Intergenic	*MIR3201*
2	*rs7563259*	A	0.382	1.566	1.299–1.889	4.692	2.70 × 10^–6^	Intergenic	*RNU5E-7P*
10	*rs74743572*	G	0.292	1.605	1.317–1.957	4.689	2.74 × 10^–6^	Intron	*RP11-383C5.4*
12	*rs17837082*	A	0.124	1.882	1.444–2.453	4.679	2.88 × 10^–6^	Intron	*RP11-121E16.1*
2	*rs7602146*	G	0.382	1.563	1.296–1.886	4.673	2.97 × 10^–6^	Intergenic	*RNU5E-7P*
5	*rs1469393*	T	0.031	3.020	1.899–4.802	4.671	3.00 × 10^–6^	Upstream transcript	*FAM81B*
2	*rs6725714*	G	0.382	1.563	1.296–1.885	4.670	3.01 × 10^–6^	Intergenic	*RNU5E-7P*
10	*rs79219793*	G	0.334	1.576	1.301–1.910	4.652	3.29 × 10^–6^	Intron	*RP11-383C5.4*
2	*rs10929394*	C	0.383	1.557	1.291–1.877	4.630	3.67 × 10^–6^	Intergenic	*RNU5E-7P*
10	*rs57227860*	G	0.464	0.644	0.533–0.778	–4.568	4.93 × 10^–6^	Upstream transcript	*CDH23*
19	*rs8104061*	A	0.174	1.704	1.355–2.143	4.561	5.10 × 10^–6^	Intron	*ZNF8*
9	*rs2765969*	T	0.337	1.552	1.285–1.875	4.557	5.18 × 10^–6^	Intergenic	*RPL4P5*
19	*rs112884174*	C	0.174	1.703	1.355–2.141	4.557	5.19 × 10^–6^	Intron	*ZNF8*
19	*rs10421600*	A	0.174	1.702	1.354–2.140	4.552	5.30 × 10^–6^	Intron	*ZNF8*
19	*rs10420807*	C	0.174	1.702	1.354–2.140	4.552	5.30 × 10^–6^	Intron	*ZNF8*
19	*rs7253055*	A	0.174	1.700	1.352–2.138	4.543	5.56 × 10^–6^	Intron	*ZNF8*
3	*rs144605956*	G	0.016	5.094	2.522–10.29	4.539	5.64 × 10^–6^	Intergenic	*RP11-654C22.2*
12	*rs74497942*	A	0.131	1.836	1.412–2.387	4.538	5.68 × 10^–6^	Intron	*RP11-121E16.1*
9	*rs2986259*	C	0.350	1.543	1.278–1.862	4.508	6.55 × 10^–6^	Intergenic	*KDM4C*
11	*rs2292731*	T	0.385	0.637	0.523–0.775	–4.504	6.68 × 10^–6^	Intergenic	*MMP20*
19	*rs57831195*	C	0.070	2.130	1.532–2.961	4.498	6.85 × 10^–6^	Intergenic	*RFPL4A*
12	*rs61432389*	C	0.131	1.820	1.402–2.362	4.495	6.97 × 10^–6^	Intron	*RP11-121E16.1*
2	*rs6754791*	T	0.381	1.536	1.273–1.853	4.486	7.26 × 10^–6^	Intergenic	*RNU5E-7P*
2	*rs6726202*	G	0.381	1.536	1.273–1.852	4.484	7.32 × 10^–6^	Intergenic	*RNU5E-7P*
2	*rs4669006*	A	0.374	1.538	1.274–1.857	4.480	7.46 × 10^–6^	Intergenic	*RNU5E-7P*
12	*rs17837077*	A	0.130	1.823	1.402–2.371	4.477	7.56 × 10^–6^	Intron	*RP11-121E16.1*
2	*rs4566338*	C	0.372	1.536	1.273–1.855	4.469	7.86 × 10^–6^	Intergenic	*RNU5E-7P*
1	*rs34207591*	G	0.090	1.990	1.471–2.692	4.466	7.97 × 10^–6^	Upstream transcript	*SETDB1*
1	*rs71624514*	T	0.090	1.990	1.471–2.692	4.466	7.97 × 10^–6^	Upstream transcript	*SETDB1*
12	*rs7488720*	G	0.131	1.815	1.397–2.359	4.461	8.17 × 10^–6^	Intron	*RP11-121E16.1*
12	*rs75289086*	C	0.133	1.805	1.392–2.341	4.452	8.52 × 10^–6^	Intron	*RP11-121E16.1*
9	*rs2997570*	T	0.335	1.537	1.272–1.858	4.446	8.76 × 10^–6^	Intergenic	*KDM4C*
12	*rs78145337*	C	0.130	1.812	1.394–2.356	4.436	9.16 × 10^–6^	Intron	*RP11-121E16.1*
10	*rs17153422*	C	0.286	1.568	1.285–1.912	4.431	9.40 × 10^–6^	Intron	*RP11-383C5.3*
9	*rs2997577*	A	0.343	1.532	1.268–1.850	4.428	9.49 × 10^–6^	Intergenic	*KDM4C*
16	*rs12928443*	A	0.187	1.677	1.334–2.109	4.426	9.58 × 10^–6^	Intergenic	*CTD-2535I10.1*
12	*rs111721834*	C	0.131	1.805	1.389–2.345	4.419	9.90 × 10^–6^	Intron	*RP11-121E16.1*
9	*rs2997572*	T	0.336	1.533	1.268–1.853	4.418	9.96 × 10^–6^	Intergenic	*KDM4C*

^a^ Chromosome; ^b^ hg19, dbSNP150 version; ^c^ alternative allele; ^d^ minor allele frequency.

**Table 3 ijms-24-12266-t003:** Thirteen SNPs with asthma-related transcription factors identified using functional annotation.

CHR	SNP	CADD ^a^	DANN ^b^	Transcription Factors ^c^	Rank Score ^d^	Gene
1	*rs139189121*	11.940	0.792	AR [92]	5	*SETDB1*
1	*rs75406390*	12.280	0.836	AR [92]	5	*SETDB1*
1	*rs59024312*	8.982	0.833	FOXA1 [93], GATA3 [94,95], GATA6 [96,97]	4	*SETDB1*
19	*rs8104061*	0.751	0.718	EZH2 [98]	2b	*ZNF8*
19	*rs112884174*	3.736	0.812	EZH2 [98]	4	*ZNF8*
19	*rs260498*	10.310	0.895	POLR2A [99]	4	*ZNF8*
19	*rs11671486*	11.590	0.886	POLR2A [99]	1f	*ZNF8*
11	*rs2245803*	17.260	0.985	RXRA [100,101], FOXP1 [102], NFIL3 [103,104], RARA [105], SMAD4 [106]	3a	*MMP20*
16	*rs2532008*	0.206	0.558	RXRA [100,101], ZNF341 [107,108], PRDM10 [109,110]	2b	*ADCY9*
16	*rs2532007*	0.350	0.382	ATF3 [111,112], BCL6 [113,114], EGR1 [115,116], ELF3 [117], EP300 [118,119], FOXA1 [93], FOXA2 [94,95], FOXA3 [94,120], HDAC2 [121,122], IRF2 [123], JUND [124,125], KAT8 [126,127], NFIL3 [103,104], NFYC [128], PRDM1 [129,130], RAD21 [131,132], RARA [105], RXRA [100,101], RXRB [133], SMAD4 [106], SP1 [134], THRB [135,136], YY1 [137,138] ZEB2 [139], and USF1 [140,141]	2b	*ADCY9*
16	*rs384067*	2.631	0.677	CTCF [142,143], CREB1 [144,145]	2b	*ADCY9*
9	*rs10814023*	16.070	0.540	POLR2A [99], CTCF [142,143]	3a	*DOCK8*
9	*rs10758219*	13.540	0.851	POLR2A [99], CTCF [142,143]	3a	*DOCK8*

^a^ The CADD PHRED score 10 means risk of top rank 10%, 20 means risk of top rank 1%; ^b^ the DANN score ranges from 0 to 1, where 1 represents the highest possibility for pathogenicity; ^c^ asthma-related transcription factor that controls the rate of transcription of genetic information from DNA to messenger RNA by binding to a specific DNA sequence; ^d^ scores range from 1 to 7, with lower values indicating that the variant is more likely to be located in the functional region.

**Table 4 ijms-24-12266-t004:** Summary of the RegulomeDB category descriptions.

RegulomeDB Category	Category Description
Likely to affect binding and linked to expression of a gene target	
1a	eQTL ^a^ + TF ^b^ binding + matched TF motif + matched DNase ^c^ footprint + DNase peak
1b	eQTL + TF binding + any motif + DNase footprint + DNase peak
1c	eQTL + TF binding + matched TF motif + DNase peak
1d	eQTL + TF binding + any motif + DNase peak
1e	eQTL + TF binding + matched TF motif
1f	eQTL + TF binding/DNase peak
Likely to affect binding	
2a	TF binding + matched TF motif + matched DNase footprint + DNase peak
2b	TF binding + any motif + DNase footprint + DNase peak
2c	TF binding + matched TF motif + DNase peak
Less likely to affect binding	
3a	TF binding + any motif + DNase peak
3b	TF binding + matched TF motif
Minimal binding evidence	
4	TF binding + DNase peak
5	TF binding or DNase peak
6	Motif hit

^a^ Expression quantitative trait locus (eQTL); ^b^ transcription factor (TF); ^c^ deoxyribonuclease (DNase).

## Data Availability

The HEXA, CAVAS, and KARE Korean Chip (KORV1.1) datasets are a part of KoGES and are available upon approval of the genome center at the Korea National Institute of Health (https://is.kdca.go.kr/ (accessed on 1 June 2023)).

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
