# Peer review of "Gene–Smoking Interaction Analysis for the Identification of Novel Asthma-Associated Genetic Factors"

_ijms, 2023, doi:10.3390/ijms241512266_

Round 1

Reviewer 1 Report

In their manuscript, Cha and Choi tried to explore gene-environment interaction analysis following smoking as a confounder to find novel genes related to asthma. Beside the known main genes founds, the authors found SETDB1 and ZNF8 genes as new targets. Their analysis is purely bio informational based on Korean cohorts.

I have the following comments:

Major Comments

-       In the introduction, the authors mentioned that asthma is caused by infection agents. This wrong and needs to be corrected. This raises questions about their knowledge about asthma and various effector mechanisms of Asthma.

-       The title is misleading. The authors focused on smoking only. It must be mentioned clearly in the title.

-       In their analysis they have missed many Asthma related genes including Notch4 for examples. Can they comment on that? I think it is very important to introduce more Asthma related genes in their cohorts as a proof of principle of their pipeline analysis.

-       Their explanation of the role of the two novel genes seems very mechanical an theoretical. I don’t think real experiments to verify the genes can be done at this stage, but it would be nice

English Language is good. Minor correction are needed

Author Response

Dear Reviwer,

I am wish to submit our revised manuscript, titled “Gene–smoking interaction analysis for the identification of novel asthma-associated genetic factors” (Manuscript ID: ijms-2510752) for consideration for publication in the International Journal of Molecular Sciences.

We want to thank the reviewers for their positive and constructive comments. We are happy to let you know that we have addressed all the reviewer’s concerns regarding the previous version of the manuscript. As a result, we believe that the revised manuscript and supplements have considerably improved. Our point-by-point responses to the reviewer’s comments are detailed below this letter.

Thank you for considering this paper for publication, and I look forward to hearing from you.

Sincerely,

Sungkyoung Choi

Author Response

Dear Reviewer,

I am wish to submit our revised manuscript, titled “Gene–smoking interaction analysis for the identification of novel asthma-associated genetic factors” (Manuscript ID: ijms-2510752) for consideration for publication in the International Journal of Molecular Sciences.

We want to thank the reviewers for their positive and constructive comments. We are happy to let you know that we have addressed all the reviewer’s concerns regarding the previous version of the manuscript. As a result, we believe that the revised manuscript and supplements have considerably improved. Our point-by-point responses to the reviewer’s comments are detailed below this letter.

Thank you for considering this paper for publication, and I look forward to hearing from you.

Sincerely,

Sungkyoung Choi

Reviewer 3 Report

The authors investigated asthma-associated  gene–environment interactions at the level of single nucleotide polymorphisms,  genes, and gene sets  identifying  two novel genes (SETDB1 and ZNF8) and five previously  reported genes associated with increased asthma risk.

In my opinion the manuscript is a little hard to follow in certain sections, in particular  section 4.5 should be made more more understandable and relative table 4 in which some acronyms are not explicited (for example eQTL) should be edited and better explained.

Furthermore, it is not clear if  the found SNPs are associated with an increased risk of developing asthma both in presence of smoke or also in absence, it should be better explained.

Moreover , I wonder if the found SNPs are within a coding sequence of a gene, within an intronic region, or in an intergenic region, influencing the transcription level of the genes.

Minor change to do :

 Line 88: change AGE with Age

Author Response

(The authors gave the same response as above.)

Round 2

Reviewer 1 Report

The authors improved it and i recommed to accept it

Reviewer 2 Report

The 2nd manuscript has been revised accordingly. This study is worthy of publication in this journal.

Reviewer 3 Report

  The authors have responded to the reviewer's requests and considering the new version, the manuscript can be accepted in the present form.